# Radiation-Induced Retinopathy and Optic Neuropathy after Radiation Therapy for Brain, Head, and Neck Tumors: A Systematic Review

**DOI:** 10.3390/cancers15071999

**Published:** 2023-03-27

**Authors:** Buket Kinaci-Tas, Tanja Alderliesten, Frank D. Verbraak, Coen R. N. Rasch

**Affiliations:** 1Department of Radiation Oncology, Leiden University Medical Centre, 2300 RC Leiden, The Netherlands; 2Department of Ophthalmology, Amsterdam University Medical Centers, Location VU Medical Center, 1081 HV Amsterdam, The Netherlands

**Keywords:** radiation retinopathy, radiation optic neuropathy, brain tumors, head and neck tumors, radiation therapy

## Abstract

**Simple Summary:**

Retinopathy and optic neuropathy are well-known, severe ocular complications in patients undergoing radiation therapy for brain, head, and neck cancer. However, little is known about the prevalence and dose–response relationship of retinopathy and optic neuropathy in these patients. More knowledge about the prevalence and dose–response relationship may contribute to developing a high-precision radiation therapy approach.

**Abstract:**

Background: Patients with brain, head, and neck tumors experience a decline in their quality of life due to radiation retinopathy and optic neuropathy. Little is known about the dose–response relationship and patient characteristics. We aimed to systematically review the prevalence of radiation retinopathy and optic neuropathy. Method: The primary outcome was the pooled prevalence of radiation retinopathy and optic neuropathy. The secondary outcome included the effect of the total radiation dose prescribed for the tumor according to the patient’s characteristics. Furthermore, we aimed to evaluate the radiation dose parameters for organs at risk of radiation retinopathy and optic neuropathy. Results: The pooled prevalence was 3.8%. No retinopathy was reported for the tumor’s prescribed dose of <50 Gy. Optic neuropathy was more prevalent for a prescribed dose of >50 Gy than <50 Gy. We observed a higher prevalence rate for retinopathy (6.0%) than optic neuropathy (2.0%). Insufficient data on the dose for organs at risk were reported. Conclusion: The prevalence of radiation retinopathy was higher compared to optic neuropathy. This review emphasizes the need for future studies considering retinopathy and optic neuropathy as primary objective parameters.

## 1. Introduction

Radiation therapy is a mainstay treatment for various brain, head, and neck tumors. Due to the complex anatomical relationship between the tumor and critical structures such as the retina, optic nerve, and chiasm, it is nearly impossible to prevent partial irradiation of these structures [1,2,3,4]. Although remarkable progress has been made with respect to the visualization of tumors for treatment planning purposes and the application of highly accurate radiation dose delivery, the retina, optic nerve, and chiasm will inevitably receive a radiation dose that exceeds its normal tissue tolerance. Depending on the extent of the involvement of the retina, optic nerve, and chiasm in the radiation field, variable acute and late ocular damage may occur. Radiation retinopathy and radiation optic neuropathy are relatively common late ocular toxicities of radiation therapy for tumors arising in the brain, head, and neck [1,2,5,6,7,8].

Both retinopathy and optic neuropathy have important clinical consequences that may deteriorate a patient’s quality of life [9,10,11]. Patients mainly present visual symptoms, including partial or complete loss of vision and visual field defects [12,13]. Retinopathy is caused by a slow, progressive vasculopathy after radiation exposure, potentially leading to microaneurysms, dot-blot hemorrhages, capillary closure, exudates, neovascular proliferation, macular edema, and neuroretinal degeneration [14]. Optic neuropathy is characterized by vessel damage and occlusion followed by optic nerve atrophy and loss of retinal nerve fibers [15,16,17].

There is a lack of accurate estimation for retinopathy and optic neuropathy prevalence in patients receiving radiation therapy for brain, head, and neck tumors. We aim to systematically review the prevalence of retinopathy and optic neuropathy among existing studies and evaluate the impact of both radiation dose parameters and the patient’s characteristics (i.e., age, sex, comorbidities, chemotherapy, and medical history) on retinopathy and optic neuropathy prevalence.

## 2. Materials and Methods

### 2.1. Search Strategy

We conducted this systematic review according to the protocol of the Preferred Reporting Items for Systematic Reviews and Meta-Analysis [18]. A clinical librarian conducted the literature search in the PubMed, Embase, and Cochrane Library databases on 11 July 2022. All publications published in the English language were selected for screening based on the following search terms: “radiation retinopathy”, “optic neuropathy”, and “brain tumors” and/or “head and neck tumor”. A detailed search syntax is demonstrated in Appendix B.

### 2.2. Eligibility Criteria and Study Selection Process

Two reviewers independently selected all relevant articles. First, they eliminated duplicates. Subsequently, they excluded irrelevant articles by title and abstract. Finally, the remaining articles were reviewed and selected for eligibility based on reading the full-text manuscripts. Studies were considered eligible when patients underwent radiation therapy for brain, head, and/or neck cancer and if the prevalence of retinopathy and/or optic neuropathy was reported as an outcome measurement. Furthermore, the diagnosis of retinopathy and optic neuropathy had to be confirmed by ophthalmologic examination. Reviews, case reports, comments, or letters were excluded, as were articles published before 1980. The reviewers applied no restrictions to the study design or type of radiation therapy. The two reviewers resolved any discrepancies through discussion until they reached a consensus.

### 2.3. Data Extraction

The following data were extracted from the included articles: study design, year of publication, total number of patients, type of tumor, radiation therapy technique, radiation tumor dose; the minimum, maximum, and the mean total dose prescribed to the optic nerve and retina; fractionation schedule, the time between radiation therapy and the onset of retinopathy and optic neuropathy; age, sex, and chemotherapy use. Next, we summarized and graphically displayed the extracted data in tables and scatterplots.

### 2.4. Outcome Definition

The primary outcome was the pooled prevalence and confidence intervals of retinopathy and optic neuropathy, as confirmed by ophthalmologic examination. The secondary outcome was the total dose prescribed for the tumor, including the minimum, maximum, and mean doses received by the optic nerve, retina, and chiasm. Furthermore, we aimed to evaluate the effect of both radiation dose parameters (i.e., total radiation dose prescribed for the tumor and the doses received by the retina, optic nerve, and chiasm) and the patient’s characteristics (i.e., age, sex, comorbidities, chemotherapy, and medical history) on retinopathy and optic neuropathy prevalence. We compared the data using the chi-square test in univariate analysis. A two-sided *p*-value < 0.05 was considered statistically significant.

## 3. Results

Our literature search on 11 July 2022 yielded 2748 articles, which we screened for inclusion after duplicates were discarded. We determined 288 relevant articles after the title and abstract screening. As shown in Figure 1, 97 articles were excluded due to the following reasons: case reports (*n* = 53), not written in English (*n* = 20), published before 1980 (*n* = 13), reviews (*n* = 102), and no clear description of the prevalence of retinopathy and/or optic neuropathy (*n* = 16). Furthermore, we excluded six articles reporting comments on published articles regarding radiation retinopathy. We included 78 articles in our final analysis.

In total, 74 studies described a retrospective study design, and 4 studies described a prospective study design. Four studies distinguished two groups in the study population according to the tumor type or radiation therapy technique [19,20,21,22]. Therefore, we divided these studies and analyzed the subgroups as separate study groups. Of these included studies, 21 mentioned the prevalence of retinopathy and/or optic neuropathy after radiation therapy for pituitary adenomas, 14 for sinus malignancies, and 10 for nasopharynx carcinoma. The remaining studies mentioned the prevalence of retinopathy and/or optic neuropathy after radiation therapy for meningioma, cephalic malignancies, chordoma, chondrosarcoma, glioma, and paranasal and sellar malignancies. The radiation therapy technique in the studies consisted of external beam radiation therapy, gamma knife radiosurgery, stereotactic radiosurgery, and proton radiation therapy. The studies were conducted between 1983 and 2021. A total of 11,279 patients were included, with a mean age of 51 years (range, 18–65 years) and a mean follow-up of 57 months (range, 11–144 months). The pooled prevalence of retinopathy and optic neuropathy was 3.8%. Figure 2 and Figure 3 show data regarding the total radiation dose prescribed for the tumor for retinopathy and optic neuropathy.

### 3.1. Retinopathy and Radiation Dose Parameters

In total, 29 studies reported the prevalence of retinopathy, including 2458 patients [1,7,21,22,23,24,25,26,27,28,29,30,31,32,33,34,35,36,37,38,39,40,41,42,43,44]. Of these 29 studies, 27.6% were sinonasal tumors, 21% were nasopharyngeal tumors, 13.8% were maxillary sinus carcinoma, 10% were optic nerve sheath meningioma, 6.9% were head/neck cancers, and 3.4% consisted of skull base meningioma, cavernous malformation, and olfactory tumors. A total of 108 patients were diagnosed with retinopathy, with a prevalence of 6% CI 95% = [3.0–11.0%] (Figure 4). The diagnosis time was reported in 20 studies with a median of 39 months (range, 8–111 months) following irradiation [1,7,21,23,26,27,29,31,34,35,36,37,38,39,40]. Radiation retinopathy was observed at a prescribed dose to the tumor > 50 Gy (Figure 2). No significant association was found between the tumor’s prescribed dose and radiation retinopathy prevalence (*p* = 0.714). Of these 29 studies, 4 studies reported the prescribed dose to the retina [1,26,33,39], 6 studies reported the prescribed dose to the optic nerve [27,33,44,45,46], and 5 studies reported the prescribed dose to the chiasm [27,28,33,38]. No significant association was found between retinopathy prevalence and the dose received by the retina (*p* = 0.798) and optic nerve (*p* = 0.366). A significant association was found between the dose received by the chiasm and the prevalence of radiation retinopathy (*p* = 0.009).

### 3.2. Optic Neuropathy and Radiation Dose Parameters

In total, 70 studies reported the prevalence of optic neuropathy, totaling 10,685 patients (Figure 5) [1,15,19,20,21,22,26,28,31,32,33,35,36,38,39,40,42,43,45,48,50,51,52,53,54,55,56,57,58,59,60,61,62,63,64,65,66,67,68,69,70,71,72,73,74,75,76,77,78,79,80,81,82,83,84,85,86,87,88,89,90,91,92]. In these 70 studies, 28.6% were pituitary adenomas, 14% were sinonasal tumors, 11.4% were meningiomas, 6% were nasopharyngeal tumors, 7.1% were chordoma/chondrosarcoma or cavernous malformations, 4.2% were optic nerve sheath meningiomas, and 1.4% consisted of cephalic/glioma/(para)sellar tumors. Optic neuropathy was observed in 319 patients (prevalence = 2.0% [CI_95%_ = 0.028–0.034]). The diagnosis time of optic neuropathy was reported in 35 studies, with a mean of 36 months following irradiation (range, 3–108 months) [1,19,22,26,28,35,39,40,41,42,43,45,49,51,52,54,55,59,60,61,62,65,71,73,75,77,79,81,85,87,89]. Optic neuropathy prevalence was observed at prescribed doses of >50 Gy and <50 Gy at 4.5% and 1.7%, respectively. This difference was significant (*p* = 0.001) (Figure 6). A significant association was found between the prescribed dose for the tumor and optic neuropathy prevalence (*p* < 0.0001).

The studies included in the analysis reported limited information on the doses received by the retina, optic nerve, and chiasm. Therefore, we performed a subgroup analysis on studies that reported the prevalence of optic neuropathy and/or retinopathy with organs at risk doses. These studies did not include organs with risk doses as a primary endpoint but collected those details and analyzed which dose led to optic neuropathy or retinopathy. The prevalence of optic neuropathy and the doses received by the retina, optic nerve, and/or chiasm were reported in 33 of the 70 studies. Of these 33 studies, 10 reported the dose received by the retina, 26 reported the dose received by the optic nerve, and 24 reported the dose received by the chiasm. A significant association was found between the prevalence of optic neuropathy and the dose received by the retina (*p* = 0.009 [CI_95%_ = 0.0164–0.01140]), the optic nerve (*p* = 0.001 [CI_95%_ = 0.00146–0.0585]), and the chiasm (*p* = 0.03 [CI_95%_ = 0.0047–0.0742]).

Information regarding the radiotherapy technique per study, the mean total dose and dose per fraction is reported in the Appendix A.

### 3.3. Data concerning Patients’ Characteristics and Prescribed Doses for the Retina, Optic Nerve, and Chiasm

We found limited reports on the characteristics of the patients diagnosed with retinopathy and/or optic neuropathy. Therefore, it was impossible to analyze the effects of the patient’s characteristics on retinopathy and optic neuropathy prevalence. It was impossible to address the severity of retinopathy and optical neuropathy as no case descriptions were reported.

## 4. Discussion

Retinopathy and optic neuropathy are well-known, late ocular complications in patients undergoing radiation therapy for brain, head, and neck tumors. However, little is known about the dose–response relationship between retinopathy and optic neuropathy in these patients. More knowledge about the prevalence, severity, and dose–response relationship may contribute to developing a high-precision radiation therapy approach for these patients. For this reason, in this systematic review, we analyzed the prevalence and the severity of retinopathy and optic neuropathy in patients undergoing radiation therapy for brain, head, and neck tumors. Subsequently, we aimed to evaluate the predictive value of radiation parameters on the prevalence and severity of retinopathy and optic neuropathy.

### 4.1. Prevalence of Retinopathy

The pooled prevalence of retinopathy and optic neuropathy was 3.8%. Retinopathy prevalence was 6%, which was remarkably less than reported by one study, observing 63.6 and 36.3% in patients who underwent radiation therapy for paranasal sinus and nasopharyngeal carcinomas, respectively [21]. Another study reported a prevalence of 70% for retinopathy in patients who underwent radiation therapy for nasal cavity carcinoma [38]. This result can be explained by the fact that the radiation dose received by the retina, optic nerve, and chiasm is higher for sinus and nasopharyngeal carcinomas than that caused by the irradiation of tumors further from the retina, optic nerve, and chiasm. Furthermore, the total number of included patients in these studies who underwent an ophthalmologic examination was 11 and 10. These results contrasted other studies included in this systematic review where no routine ophthalmologic examination was performed or only performed when patients reported visual symptoms. Retinopathy was mainly observed at a prescription dose of >50 Gy. It may be that an ophthalmological examination for a radiation dose < 60 Gy is not performed due to the absence of subjective visual symptoms. Therefore, possible eye damage could not be detected and was not reported. Another reason could be that the radiation dose received by the eye was too low to cause detectable damage.

### 4.2. Prevalence of Optic Neuropathy

Regarding optic neuropathy, the prevalence in this systematic review was 2.0%. A prevalence of 2.3 and 0.9% for patients treated with pencil beam scanning proton therapy (PBSPT) was reported for chondrosarcoma and chordoma, respectively [93]. Another group reported a prevalence of 2.2% in patients who underwent a combination of photon and proton therapy for chordoma or chondrosarcoma [94]. In a study that evaluated the long-term results of patients who received spot-scanning proton therapy for intracranial meningioma, the prevalence of optic neuropathy was 7.7% [95]. In a study that focused on patients who underwent PBSPT for skull base and head and neck tumors, optic neuropathy was observed for 6.5% of patients [71]. However, the authors only included patients who received ≥45 Gy to their optic nerve and/or chiasm. These patients were more likely to develop optic neuropathy. Although no threshold dose for the development of optic neuropathy has been established, in this systematic review, optic neuropathy prevalence was higher at a delivered dose > 50 Gy (3.2%) than at <50 Gy (1.1%). Optic neuropathy incidences ranging from 0% for <50 Gy to 16% for >70 Gy have been reported [54]. Similar findings were reported in a study where the risk of optic neuropathy was between 3 and 7% for patients who received doses of 55–60 Gy, and 7–20% for doses > 60 Gy [2]. Currently, the generally accepted threshold to limit the risk of retinopathy is 55 Gy in 1.8–2.0 Gy fractions [96,97]. We found a higher prevalence when >50 Gy was delivered to the optic nerve than <50 Gy. However, another study reported that a high dose administered to the anterior optic pathways increased the risk of retinopathy, and patients with tumors near the anterior optic pathways have a significantly higher risk of developing retinopathy. This result may be due to the direct damage the tumor caused to the nerve/vascular tissue of the anterior optic pathways, resulting in compression and reduced radiation tolerance [71]. In this systematic review, the highest observed prevalence of retinopathy was 63.3% and 70% for nasopharyngeal and posterior nasal tumors, respectively [19,39]. These are close to the anterior optic pathways and may induce nerve/vascular damage to the anterior optic pathways before radiation therapy. Another factor that merits attention is that this review reports that cases of radiation retinopathy are rare at doses below 50 Gy, while radiation optic neuropathy can occur at doses < 50 Gy. The optic nerve is a critical structure with a limited ability to repair itself. If there have been pre-existing factors such as compression and dysfunction of the optic nerve prior to radiation therapy, this may contribute to an increase in the susceptibility of radiation neuropathy after radiation therapy [89]. In this review, 59% of the studies reporting neuropathy at doses less than 50 Gy consist of pituitary tumors. The remaining 41% of the studies consist of cavernous sinus meningioma and craniopharyngioma. The most common neuro-ophthalmological symptom of pituitary tumors prior to radiation therapy is impaired vision. This is caused by the compression of the optic chiasm, and the visual defects depend on the degree and site of optic nerve compression [98].

Using different constraints, such as different imaging modalities (OCT/OCT-angiography or UWFA) or different follow-up schedules, may be appropriate for patients with tumors close to the anterior optic pathway. Additionally, in addition to regular ophthalmological evaluation, a baseline assessment to identify structural damage in the retina and optic nerve, such as visual field testing, visual acuity testing, and other relevant exams, including OCT and OCT-angiography, can provide a comprehensive understanding of the patients’ baseline vision and can assist in monitoring changes over time. Next to the assessment of the structural abnormalities, the assessment of functional abnormalities could also contribute to this. Electrophysiology tests, such as full-field electroretinography, pattern electroretinography, and visual evoked potential, can provide valuable information about the functional status of the retina and optic nerve. These evaluations can help in making informed decisions about treatment and management for these patients.

It is important to mention that in most included studies, the incidence of retinopathy and optic neuropathy was not the primary objective of the studies and, therefore, not defined in detail. The included studies mainly mentioned retinopathy and optic neuropathy in patients with visual complaints. Furthermore, the studies’ follow-up time and extent of ophthalmological examinations differ significantly. Consequently, the presence of subclinical damage was not evaluated, which may have led to a considerable underreporting of retinopathy and optic neuropathy in this population. A routine follow-up examination and standardized classification would be useful for reporting retinopathy and optic neuropathy, predicting the prognosis more precisely, and starting timely treatment if needed. Implementing such a classification system should become part of the protocol in the follow-up examination of patients undergoing radiation therapy for brain, head, and neck tumors. Nevertheless, so far, no widely used classification system has been described to accurately assess retinopathy or optic neuropathy. Generally, retinopathy is first assessed by fundus examination in which dot-blot hemorrhages, microaneurysms, and exudates can be seen, depending on the findings of the fundus examination and other diagnostic tools. However, before these clinical features manifest, the underlying vascular damage would have already occurred and cannot be detected with fundus examination only. The main diagnostic tools to assess the underlying damage of retinopathy and optic neuropathy consist of fundus photography, fluorescein angiography (FA), OCT, and OCT-angiography (OCT-A). Fundus photography can detect major abnormalities such as cotton-wool spots, exudates, and intraretinal blot hemorrhages. FA can detect vasculopathy in more detail and more clearly (e.g., microaneurysms, areas of non-perfusion, and (neo)vascular leakage) than fundus photography. With OCT examination, the presence or absence of intraretinal or subretinal macular fluid and optic disc edema can be assessed on a micrometer scale. It can also detect neurodegenerative changes in the macular area and around the optic disc (the thickness of the peripapillary retinal nerve fiber layer) [99,100]. OCT-A can visualize and quantify the smallest capillaries in the macula and around the optic disc [101,102,103].

Several studies have aimed to create a classification system for retinopathy. A four-stage classification of retinopathy based on a combination of FA and slit lamp examination was developed [104]. The higher the stage in which the patient is classified, the higher the risk of vision loss in the affected eye. Another study expanded this classification system by adding OCT to identify macular edema [105]. Furthermore, one study proposed a classification for retinopathy in which OCT-A is used to detect retinal vascular changes [106]. They reported that OCT-A might detect retinopathy prior to changes seen on OCT alone. A recent development in imaging techniques is ultra-wide-field fluorescein angiography (UWFA), which evaluates nearly the entire fundus in a single image. A grading scheme in which UWFA describes the severity of retinopathy and predicts the progression of retinopathy in patients undergoing radiation therapy for uveal melanoma has also been developed. UWFA may also become a potential imaging technique for detecting periretinal damage, thereby improving classifications for retinopathy [107]. UWFA is a potent ophthalmologic imaging technology that can potentially screen for retinopathy because it provides images that capture up to 200° of the retina in a single capture. The most well-studied and published pathology in the era of UWFA imaging is diabetic retinopathy (DR). The clinical findings of the retina in DR are similar to those of radiation-induced retinopathy. UWFA imaging reportedly has high sensitivity (84–94%) and specificity (90–100%) for screening DR. Identical findings can be expected for detecting retinopathy. Imaging the peripheral retina at 200° provided more insight into the peripheral retinal involvement in DR than was known before. An increasing number of studies have emphasized the role of peripheral retinal pathology in early disease detection and determining DR progression [108].

Regarding optic neuropathy, a grading system characterized by five grades (grades 0–4) based on OCT-A findings has been proposed [109]. Patients were assigned a grade from 0 to 4 according to vascular abnormalities and the size of the affected area in the radial peripapillary capillary plexus (RPCP). The more vascular abnormalities and the larger the area in the RPCP, the higher the grade assigned to the patients. No other attempts have been made to grade retinopathy and optic neuropathy. Unfortunately, attempts to create a classification system for retinopathy and optic neuropathy have not resulted in widespread use. A classification system may help predict the severity of retinopathy and optic neuropathy and adjust the treatment if needed. Furthermore, it will provide a common language and frame of reference for clinical practice and research.

Another issue meriting attention is that subclinical damage detection may be relevant to the timely treatment and prevention of permanent visual loss. Early treatment with anti-vascular endothelial growth factor (anti-VEGF) or steroid-based therapies for radiation-induced macular edema will improve visual acuity [110,111,112,113,114]. Preventive treatments with scatter laser and anti-VEGF therapy will reduce the risk of complications related to neovascularization and improve visual acuity in the long term following radiation therapy [5,115,116,117].

### 4.3. Limitations

The first limitation of this systematic review was the lack of precise information about the total doses that the retina and optic nerve received. As the included studies were mainly focused on treatment effects and survival, they reported limited information on side effects, such as retinopathy and optic neuropathy. Radiation dose parameters on the retina, optic nerve, and chiasm were rarely a primary outcome parameter in the included studies. Moreover, the characteristics of patients with confirmed retinopathy or optic neuropathy were not or incompletely reported. Therefore, no correlation analysis could be performed on the prevalence and severity of retinopathy and optic neuropathy and patient characteristics, such as age, sex, comorbidities, chemotherapy, and medical history. Furthermore, nearly all the studies were missing a clear description of the diagnostic approach used for retinopathy and optic neuropathy. Furthermore, a clear description of the follow-up ophthalmologic examination was lacking regardless of a retinopathy or optic neuropathy diagnosis. We could not retrieve information on when or if patients underwent an ophthalmological examination. Information regarding the onset of the first visual symptoms was especially lacking. Therefore, an analysis of the onset of retinopathy and optic neuropathy was impossible.

### 4.4. Future Perspective

This systematic review revealed the need for more knowledge about the characteristics of patients diagnosed with retinopathy or optic neuropathy after radiotherapy for brain, head, and neck cancer. A more refined estimation, e.g., individual-based, is needed for retinopathy and/or optic neuropathy prevalence and information on subclinical damage and radiation dose to the organs at risk (i.e., retina and optic nerve). This method may increase knowledge about the tolerance dose of organs at risk. UWFA is a novel imaging technique with promising results for grading radiation retinopathy. It may enhance the understanding and management of both retinopathy and optic neuropathy. An individualized trade-off between tumor coverage and sparing organs at risk is realizable if the tolerance doses are known. Therefore, future studies are needed to determine the retina and optic nerve’s tolerance dose and their potential relationship with patient characteristics regarding retinopathy and/or optic neuropathy development.

## 5. Conclusions

In conclusion, we observed a reported prevalence rate of 6 and 2.0% for retinopathy and optic neuropathy, respectively, for patients undergoing radiation therapy for brain, head, and neck cancer. No retinopathy was reported at a prescribed dose of <50 Gy. The prevalence of optic neuropathy was higher at a prescribed dose of >50 Gy (4.5%) than at <50 Gy (1.7%). Limited studies have considered retinopathy and/or optic neuropathy as a primary objective and merely report on retinopathy and/or optic neuropathy as a side effect. Therefore, there are limited data reporting on patients’ clinical presentation and radiation dose parameters. An accepted grading scheme would be useful for identifying and staging retinopathy and optic neuropathy. It may also help identify early complications from radiation therapy. Ophthalmologic evaluation is recommended for patients receiving >60 Gy to the optic nerve and/or >50 Gy to parts of the retina. Patients who experience reduced vision following radiation therapy involving the optic nerve should be advised to seek consultation with an ophthalmologist, irrespective of the radiation dose received.

## Figures and Tables

**Figure 1 cancers-15-01999-f001:**
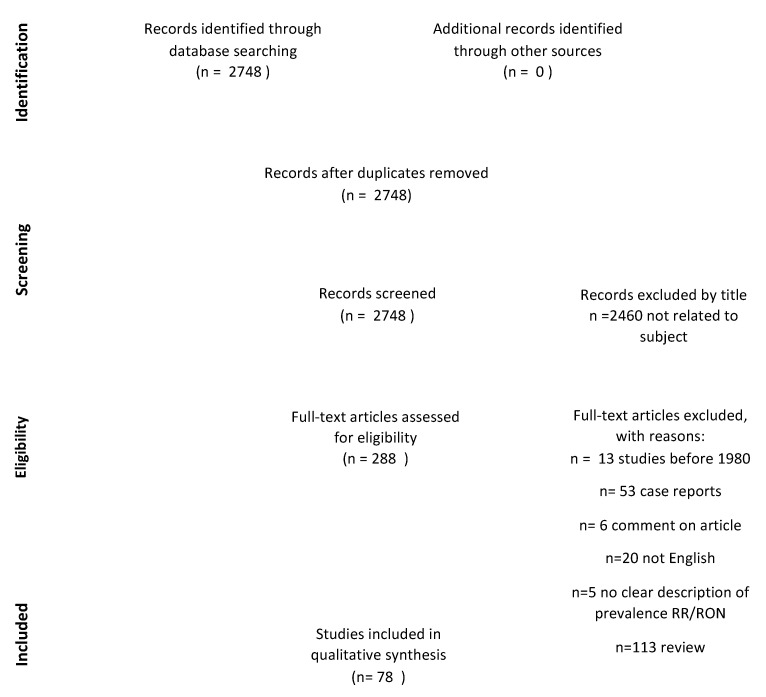
Flow diagram demonstrating the selection process of the included studies.

**Figure 2 cancers-15-01999-f002:**
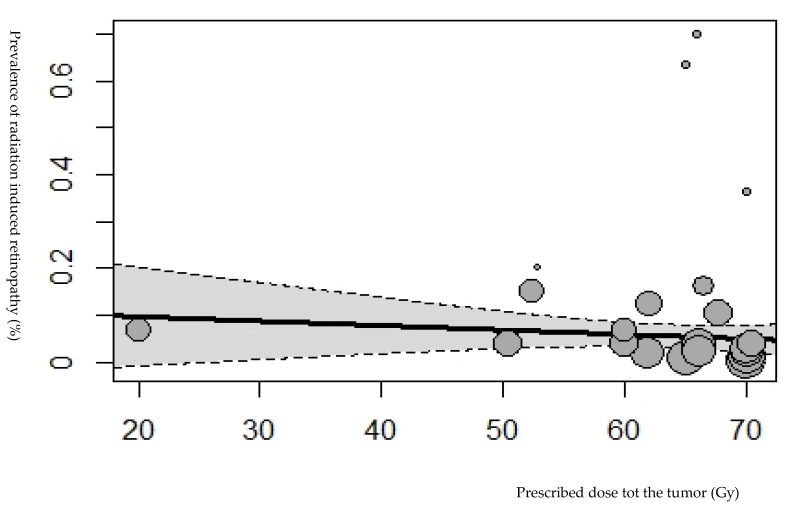
Bubble plot showing the prevalence of radiation-induced retinopathy plotted against the tumor’s prescribed dose (Gy). The size of the points is drawn proportional to the total number of patients in the studies (167). Based on the mixed-effects meta-regression model, the predicted average prevalence as a function of the tumor’s prescribed dose is also shown in the plot, with a corresponding 95% confidence interval.

**Figure 3 cancers-15-01999-f003:**
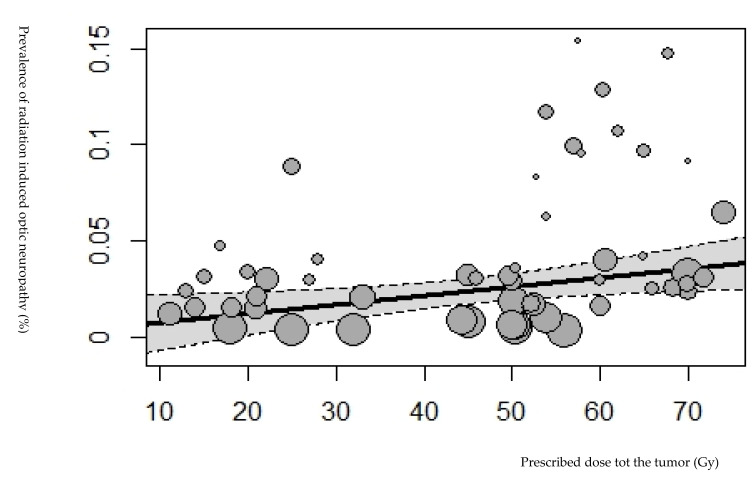
Bubble plot showing the prevalence of radiation-induced optic neuropathy plotted against the tumor’s prescribed dose (Gy). The size of the points is drawn proportional to the total number of patients in the studies. Based on the mixed-effects meta-regression model, the predicted average prevalence of radiation-induced optic neuropathy as a function of the tumor’s prescribed dose is also shown in the plot with a corresponding 95% confidence interval.

**Figure 4 cancers-15-01999-f004:**
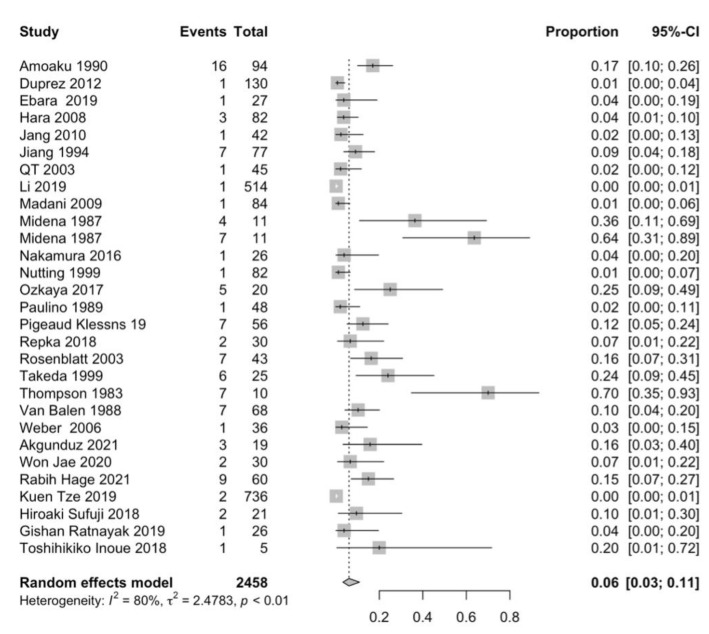
Forest plot of studies that reported radiation-induced retinopathy events [1,7,21,23,24,25,26,27,29,30,31,32,33,34,35,36,37,38,39,40,42,44,45,46,47,48,49].

**Figure 5 cancers-15-01999-f005:**
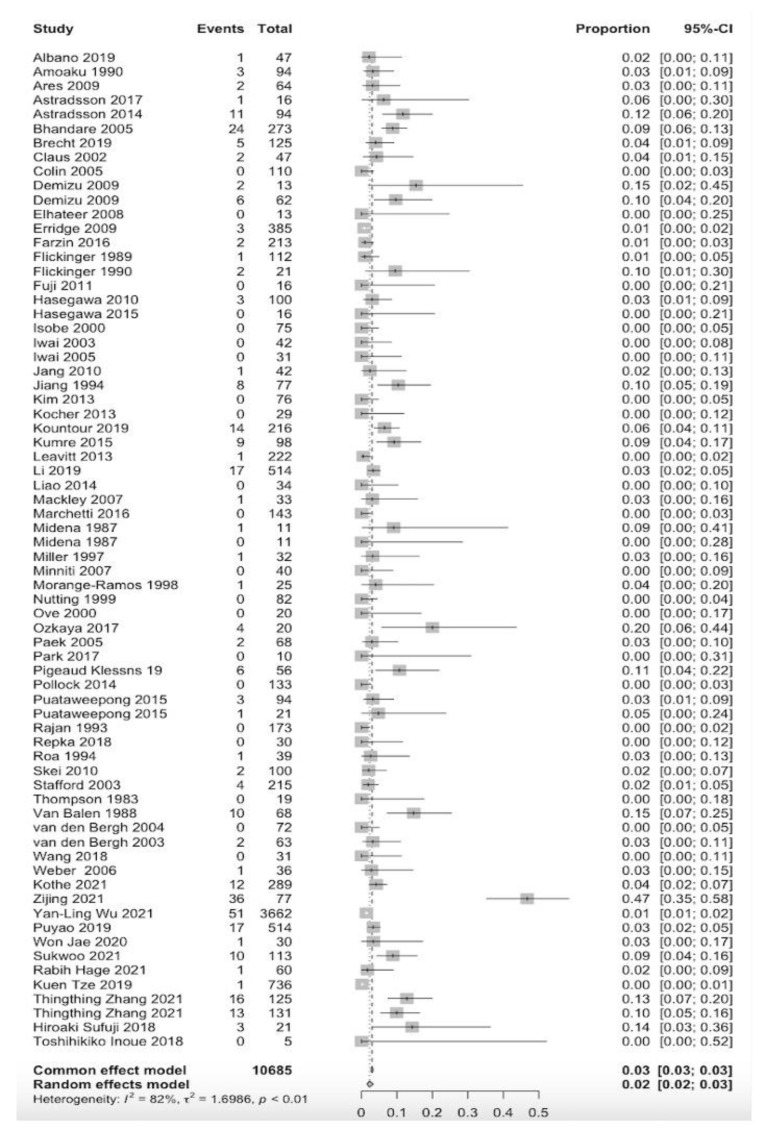
Forest plot of the studies that reported radiation-induced optic neuropathy events [1,15,19,20,21,22,26,28,31,32,33,35,36,38,39,40,42,43,44,45,48,49,50,51,52,53,54,55,56,57,58,59,60,61,62,63,64,65,66,67,68,69,70,71,72,73,74,75,76,77,78,79,80,81,82,83,84,85,86,87,88,89,90,91,92].

**Figure 6 cancers-15-01999-f006:**
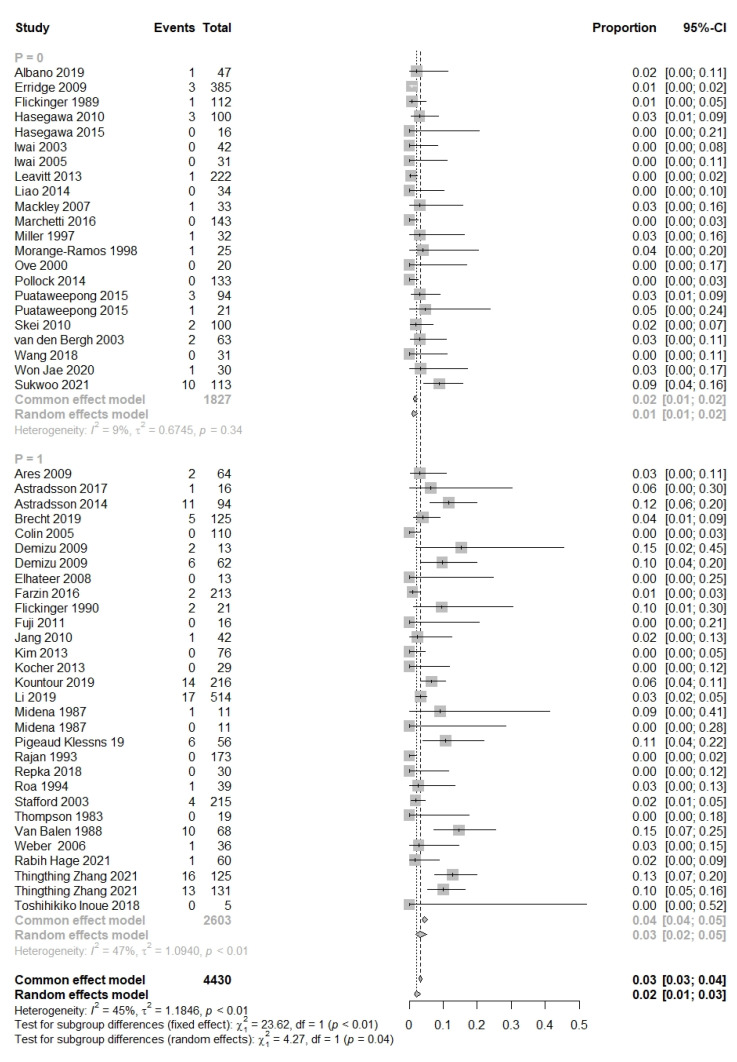
Forest plot of subgroup studies that reported radiation-induced optic neuropathy events, <50 Gy (*p*= 0) and >50 Gy (*p* = 1) [1,19,22,26,28,35,39,40,41,42,43,44,45,49,51,52,54,55,59,60,61,62,65,71,73,75,77,79,81,85,87,89].

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
