# Peer review of "Radiation-Induced Retinopathy and Optic Neuropathy after Radiation Therapy for Brain, Head, and Neck Tumors: A Systematic Review"

_cancers, 2023, doi:10.3390/cancers15071999_

Round 1

Reviewer 1 Report

The authors of this manuscript have performed a meta-analysis is manuscript describe the prevalence and dose relationship of radiation-induced retinopathy and radiation-induced optic neuropathy using both the PubMed and Embase databases. The paper is very informative; however, I have several comments and concerns:

1. The paper has numerous and significant errors of grammar, syntax, and punctuation and as well as several typographical errors (eg, line 26),. It requires editing by someone whose primary language is English.

2. Several times throughout the manuscript, the authors use the term "optic tract" to describe the anterior (pre-geniculate) visual pathway. In fact, the optic tract is the specific part of the anterior visual pathway between the optic chiasm and the lateral geniculate nucleus (LGN). The authors should substitute either the term "anterior visual pathway" if they are referring to the entire pre-geniculate pathway from the globe to the LGN or specify that they are referring to the optic nerve and chiasm; ie, not to the post-chiasmatic part of the visual pathway.

3. In general, the numbers 10 or greater should be written as numbers, not written out. There are numerous places in the manuscript that this needs to be changed (eg, lines 114 and 115).

4. Line 118. The word "chordosarcoma" should be changed to "chordoma"

5. Figure 2 Legend. The numbers "167", "168" and "169" somehow made their way into the legend. Please remove.

6. Lines 27, 97, 173, 188. The word data is plural. Please change the verb.

7. Line 271. The term "optical coherence tomography" should be added before the abbreviation "(OCT)" and "OCT-A" should be changed to "OCT angiography (OCT-A).

8. The format of the references in the reference list is not consistent. Please recheck all references to make sure that they are in the format required by the journal. 

Author Response

Dear reviewer,

We appreciate the thoughtful and thorough feedback you provided. Your attention to detail and insightful comments were invaluable in helping us to  improve our work. We have made the following necessary changes to our manuscript as a result of your feedback:

  1. The paper has numerous and significant errors of grammar, syntax, and punctuation and as well as several typographical errors (e.g., line 26). It requires editing by someone whose primary language is English.

Response author: The manuscript is now edited by MDPI’s English editing service. You can find the English Editing certificate in the attachment.

  1. Several times throughout the manuscript, the authors use the term "optic tract" to describe the anterior (pre-geniculate) visual pathway. In fact, the optic tract is the specific part of the anterior visual pathway between the optic chiasm and the lateral geniculate nucleus (LGN). The authors should substitute either the term "anterior visual pathway" if they are referring to the entire pre-geniculate pathway from the globe to the LGN or specify that they are referring to the optic nerve and chiasm; i.e., not to the post-chiasmatic part of the visual pathway.

Response author: As applicable we have substituted the term “ anterior visual pathway” with “ retina” and/or “ optic nerve”.

  1. In general, the numbers 10 or greater should be written as numbers, not written out. There are numerous places in the manuscript that this needs to be changed (e.g., lines 114 and 115).

Response author: We have written the number 10 or greater as numbers, not written out.

  1. Line 118. The word "chordosarcoma" should be changed to "chordoma"

Response author: Both words “ chordoma” and “chondrosarcoma” in line 118 are referring to studies in which patients with chordoma and chondrosarcoma underwent radiotherapy[1-3]. We prefer to use the same words as used in the original studies which we are referring to. 

  1. Figure 2 Legend. The numbers "167", "168" and "169" somehow made their way into the legend. Please remove.

Response author: The numbers “167”, “168” and “169” are now removed from the legend.

  1. Lines 27, 97, 173, 188. The word data is plural. Please change the verb.

We have changed the word data in information and/ or report.

  1. Line 271. The term "optical coherence tomography" should be added before the abbreviation "(OCT)" and "OCT-A" should be changed to "OCT angiography (OCT-A).

Response author: Line 271. The term “optical coherence tomography” is written before the abbreviation “OCT” and “OCT-A” is changed to “OCT angiography”. This is now line 263.

  1. The format of the references in the reference list is not consistent. Please recheck all references to make sure that they are in the format required by the journal.

Response author: The format of the references  list has been revised and adjusted  based on the style used by the American Chemical Society.

We hope that the revisions we have made address the concerns raised by the reviewer. We look forward to the opportunity to have our manuscript considered for publication.

Kind regards,

Buket Kinaci- Tas

References

  1. Li, P. C.; Liebsch, N. J.; Niemierko, A.; Giantsoudi, D.; Lessell, S.; Fullerton, B. C.; Adams, J.; Shih, H. A. Radiation tolerance of the optic pathway in patients treated with proton and photon radiotherapy. Radiotherapy and oncology : journal of the European Society for Therapeutic Radiology and Oncology 2019, 131, 112-119. DOI: 10.1016/j.radonc.2018.12.007
  2. Kountouri, M.; Pica, A.; Walser, M.; Albertini, F.; Bolsi, A.; Kliebsch, U.; Bachtiary, B.; Combescure, C.; Lomax, A. J.; Schneider, R.; et al. Radiation-induced optic neuropathy after pencil beam scanning proton therapy for skull-base and head and neck tumours. The British journal of radiology 2019, 20190028. DOI: 10.1259/bjr.20190028
  3. Ares, C.; Hug, E. B.; Lomax, A. J.; Bolsi, A.; Timmermann, B.; Rutz, H. P.; Schuller, J. C.; Pedroni, E.; Goitein, G. Effectiveness and safety of spot scanning proton radiation therapy for chordomas and chondrosarcomas of the skull base: first long-term report. International journal of radiation oncology, biology, physics 2009, 75 (4), 1111-1118. DOI: 10.1016/j.ijrobp.2008.12.055

Reviewer 2 Report

Overall well-written review paper on an important topic.

I have a few remarks:

1) since your search goes back to 1980, relevant radiation techniques differences will exist between the included studies. Would it be possible to show these differences (for instance 2D/3D/IMRT/VMAT/proton)

2) Besides total dose, it would also be very interesting to show dose per fraction.

3) True point made by the authors that there is a difference between patients with an optic tract that is involved or compressed by tumor and patients with a tumor with no optic tract contact. Could the authors describe whether it makes sense to use different constraints for these patients? And would the authors recommend besides follow-up ophtalmologic evaluation also baseline evaluation?

Typo's:

caption figure 2: Aagainst --> Against
line 210: posterior nasal space --> nasal cavity or paranasal sinus
line 359: missing to (to the tumor)
Conflicts of interest: Elketa --> Elekta

Author Response

Feedback Reviewer 2

We appreciate the thoughtful and thorough feedback you provided. Your attention to detail and insightful comments were invaluable in helping us to  improve our work. We have made the following changes to our manuscript as a result of your feedback:

  1. Since your search goes back to 1980, relevant radiation techniques differences will exist between the included studies. Would it be possible to show these differences (for instance 2D/3D/IMRT/VMAT/proton).
  2. Besides total dose, it would also be very interesting to show dose per fraction
    • Response author: (Q1 and Q2 combined) We have created a table that indicates which radiotherapy technique and dose per fractions was used in each study. Furthermore, per study the mean total dose and dose per fraction is shown. We have added the table as a supplementary material.  
  3. True point made by the authors that there is a difference between patients     with an optic tract that is involved or compressed by tumor and patients with a tumor with no optic tract contact. Could the authors describe whether it makes sense to use different constraints for these patients? And would the authors recommend besides follow-up ophthalmologic evaluation also baseline evaluation?
    • Response author: As we have reported in our discussion in line 280 and also mentioned in the study of Kountouri et al. We believe that patients with  tumors near the optic apparatus have an increased  risk of developing retinopathy and/or optic neuropathy. This may be explained by direct damage to nerve/vascular tissue of the anterior optic pathways by the tumor itself as a result of compression with a consequential decrease in radiation tolerance. In the study of Kountouri et al. neither Dmax or Dmean (<60 GyRBE) or dose per fraction (1.8-2.0Gy) were significantly related to the risk of RION, tumor in close to the optic apparatus did [1].

To clarify our recommendations we added the following in line 250-257:  

“Using different constraints, such as different imaging modalities ( OCT/OCT angiography) or different follow-up schedules, may be appropriate for patients with tumors close to the optic apparatus. Furthermore, in addition to regular ophthalmological evaluation, baseline evaluation such as visual field  testing, visual acuity testing, and other relevant exams like OCT and OCT-angiography can provide comprehensive understanding of the patient’s  baseline vision and can assist in monitoring changes over time. These evaluations can help in making informed decisions  about treatment and management for these patients”

We hope that the revisions we have made address the concerns raised by the reviewer and we would like to thank the reviewer once again for their valuable feedback. We look forward to the opportunity to have our manuscript considered for publication.

Kind regards,

Buket Kinaci- Tas

References

  1. Kountouri, M.; Pica, A.; Walser, M.; Albertini, F.; Bolsi, A.; Kliebsch, U.; Bachtiary, B.; Combescure, C.; Lomax, A. J.; Schneider, R.; et al. Radiation-induced optic neuropathy after pencil beam scanning proton therapy for skull-base and head and neck tumours. The British journal of radiology 2019, 20190028. DOI: 10.1259/bjr.20190028

Reviewer 3 Report

This is an interesting meta-analysis on the topic of radiation-related retinopathy and optic neuropathy. However, there are two major problems would better be addressed.

1. As mentioned in the discussion, the tumor site is a major factor to affect the incidence of these two common complication of radiation, say, NPC and para-nasal sinuses carry higher incidence as high as 36.3% and 63.6%, respectively. But in the analysis of the literature collected, the authors did not list tumor site as a factor to analyze. Also, in the estimation of the incidence, the readers would like to know how the patients groups were constitute: how much % are brain tumor? NPC? para-nasal sinuses? oral cancer? oropharyngeal cancer? laryngeal cancer?

2. In the conclusion, the authors claimed: Incidence increases with increased (tumor) pressure to the optic nerve. But I could not appreciate how this conclusion was drawn from the results and discussion.

Author Response

Dear reviewer,

We appreciate the thoughtful and thorough feedback you provided. Your attention to detail and insightful comments were invaluable in helping us to  improve our work. We have made the following changes to our manuscript as a result of your feedback:

  1. As mentioned in the discussion, the tumor site is a major factor to affect the incidence of these two common complication of radiation, say, NPC and para-nasal sinuses carry higher incidence as high as 36.3% and 63.6%, respectively. But in the analysis of the literature collected, the authors did not list tumor site as a factor to analyze. Also, in the estimation of the incidence, the readers would like to know how the patients groups were constitute: how much % are brain tumor? NPC? para-nasal sinuses? oral cancer? oropharyngeal cancer? laryngeal cancer?

Response author: We have reported the percentage of the tumors in question under the heading 3.1 Retinopathy and Radiation Dose Parameters (line 139-141) and 3.2 Optic neuropathy and radiation dose parameters (line 158-161).

  1. In the conclusion, the authors claimed: Incidence increases with increased (tumor) pressure to the optic nerve. But I could not appreciate how this conclusion was drawn from the results and discussion.

Response author: We apologize for the confusion. Indeed this statement is not covered in our results. We have only mentioned this in our discussion line 244-246. We have removed the sentence from the conclusion.

We hope that the revisions we have made address the concerns raised by the reviewer and we would like to thank the reviewer once again for their valuable feedback. We look forward to the opportunity to have our manuscript considered for publication.

Kind regards,

Buket Kinaci- Tas

Round 2

Reviewer 1 Report

The authors have revised their manuscript, particularly with respect to the grammar and syntax, and it reads much better; however, I have several further comments and concerns:

1. I previously emphasized to the authors that they were mis-using the term "optic tract" and suggested that they either use the term "anterior visual pathway" or specify retina, optic nerve, etc. They appear to have done so in some areas of the Discussion but not in all and not in the Introduction, in which the term "optic tract" continues to appear multiple times. Please revise.

2. Page 3, first paragraph, second line. The authors use the phrase "retina, chiasm, and optic nerve"; it would be more appropriate to change this to "retina, optic nerve, and chiasm"

3. Page 5, first paragraph (section 3.1), first sentence, please add a comma after the word "total"

4. Page 5, first paragraph, second sentence, please add a comma after the word "studies"

5. Page 5, first paragraph, second sentence. I do not understand the phrase "cavernosus tumor". Do you mean a tumor of the cavernous sinus or are you referring to a cavernous malformation? Please clarify.

6. Page 5, first paragraph, line 10. Please delete the word "only". Let the reader decide if the numbers of studies are "significant"

7. Page 5, last paragraph, first sentence. Please add a comma after the word "total".

8. Page 5, last paragraph, second sentence. Please change the word "From" to "In".

9. Page 6, first paragraph. The word "meningioma" is mis-spelled twice (meningeoma" and the word "cavernosus" should be changed to "cavernous"

10. Page 8, first paragraph, first sentence. This sentence does not make sense: "Information concerning the doses received by the retina, optic nerve, and chiasm were limited reported in the included  studies." I assume you mean to say that in the studies you reviewed, there was limitation information regarding the doses to these structures? Please rephrase the sentence.

11. Page 8, first paragraph, fourth sentence. Please change the phrase "nerve, chiasm, and/or retina" to "retina, nerve, and/or chiasm". Similarly, change the last sentence so that retina comes first, then nerve and then chiasm.

12. Page 8, second paragraph (section 3.3). Last sentence, please change the word "since" to "as"

13. Page 8, Discussion. As noted in #1, the term "optic tract" appears several times. Please change!

14. Page 9, Discussion, first paragraph, line 9. Please change the word "they" to "the authors".

15. Discussion. It is interesting that although the dose to the retina that causes radiation-induced retinopathy is dose-related, with virtually no cases when the dose is <50Gy, there remain cases (<2%) of radiation-induced optic neuropathy that occur with doses <50 Gy. I believe the authors should address this issue.

16. Page 9, second paragraph, line 14. The authors indicate that the assessment of the fundus is by a slit lamp fundus examination. Although it is true that many ophthalmologists use a slit lamp with a hand-held contact lens for assessment of the fundus, others use an indirect or even a direct ophthalmoscope. I'm not sure that the authors should make this so specific. The key is the examinatoin of the fundi, not what instrument you use to examine them.

17. Discussion, page 10, The authors emphasize the imaging tools (ie, OCT, OCTA, UWFA) used to improve recognition of retinopathy and optic neuropathy; however, these tools only identify structural abnormalities and not functional ones. They should add something about the use of electrophysiology (full-field ERG, PERG, VEP) to identify evidence of dysfunction, particularly with respect to radiation-induced optic neuropathy.

18. Page 10, second paragraph, sentence 3rd from the end. The authors use the abbreviation "WFA". Do they really mean WFA or should this be UWFA, as that is what they have been discussing in the paragraph?

19. Page 10, last paragraph, last sentence. Please change the word "Since" to "As" or "Because"

20. Page 11, last paragraph, last sentence. Given that radiation-induced optic neuropathy can occur (rarely) with doses less than 50 Gy, doesn't it make sense for all patients to be screened for optic neuropathy, regardless of the dose? Or at least say that patients should be warned that because radiation-induced optic neuropathy can occur at doses less than 50 Gy (1.7% is still significant if you are one of the 1.7%!), any patient who develops decreased vision after radiation therapy that encompasses the optic nerve, regardless of the dose, should be told to see an ophthalmologist immediately.

Author Response

Dear reviewer,

We appreciate the thoughtful and thorough feedback you provided. Your attention to detail and insightful comments were invaluable in helping us to  improve our work. We have made the following necessary changes to our manuscript as a result of your feedback:

  1. The paper has numerous and significant errors of grammar, syntax, and punctuation and as well as several typographical errors (e.g., line 26). It requires editing by someone whose primary language is English.

Response author: The manuscript is now edited by MDPI’s English editing service. You can find the English Editing certificate in the attachment.

  1. Several times throughout the manuscript, the authors use the term "optic tract" to describe the anterior (pre-geniculate) visual pathway. In fact, the optic tract is the specific part of the anterior visual pathway between the optic chiasm and the lateral geniculate nucleus (LGN). The authors should substitute either the term "anterior visual pathway" if they are referring to the entire pre-geniculate pathway from the globe to the LGN or specify that they are referring to the optic nerve and chiasm; i.e., not to the post-chiasmatic part of the visual pathway.

Response author: As applicable we have substituted the term “ anterior visual pathway” with “ retina” and/or “ optic nerve”.

  1. In general, the numbers 10 or greater should be written as numbers, not written out. There are numerous places in the manuscript that this needs to be changed (e.g., lines 114 and 115).

Response author: We have written the number 10 or greater as numbers, not written out.

  1. Line 118. The word "chordosarcoma" should be changed to "chordoma"

Response author: Both words “ chordoma” and “chondrosarcoma” in line 118 are referring to studies in which patients with chordoma and chondrosarcoma underwent radiotherapy[1-3]. We prefer to use the same words as used in the original studies which we are referring to. 

  1. Figure 2 Legend. The numbers "167", "168" and "169" somehow made their way into the legend. Please remove.

Response author: The numbers “167”, “168” and “169” are now removed from the legend.

  1. Lines 27, 97, 173, 188. The word data is plural. Please change the verb.

We have changed the word data in information and/ or report.

  1. Line 271. The term "optical coherence tomography" should be added before the abbreviation "(OCT)" and "OCT-A" should be changed to "OCT angiography (OCT-A).

Response author: Line 271. The term “optical coherence tomography” is written before the abbreviation “OCT” and “OCT-A” is changed to “OCT angiography”. This is now line 263.

  1. The format of the references in the reference list is not consistent. Please recheck all references to make sure that they are in the format required by the journal.

Response author: The format of the references  list has been revised and adjusted  based on the style used by the American Chemical Society.

We hope that the revisions we have made address the concerns raised by the reviewer. We look forward to the opportunity to have our manuscript considered for publication.

Kind regards,

Buket Kinaci- Tas

References

  1. Li, P. C.; Liebsch, N. J.; Niemierko, A.; Giantsoudi, D.; Lessell, S.; Fullerton, B. C.; Adams, J.; Shih, H. A. Radiation tolerance of the optic pathway in patients treated with proton and photon radiotherapy. Radiotherapy and oncology : journal of the European Society for Therapeutic Radiology and Oncology 2019, 131, 112-119. DOI: 10.1016/j.radonc.2018.12.007
  2. Kountouri, M.; Pica, A.; Walser, M.; Albertini, F.; Bolsi, A.; Kliebsch, U.; Bachtiary, B.; Combescure, C.; Lomax, A. J.; Schneider, R.; et al. Radiation-induced optic neuropathy after pencil beam scanning proton therapy for skull-base and head and neck tumours. The British journal of radiology 2019, 20190028. DOI: 10.1259/bjr.20190028
  3. Ares, C.; Hug, E. B.; Lomax, A. J.; Bolsi, A.; Timmermann, B.; Rutz, H. P.; Schuller, J. C.; Pedroni, E.; Goitein, G. Effectiveness and safety of spot scanning proton radiation therapy for chordomas and chondrosarcomas of the skull base: first long-term report. International journal of radiation oncology, biology, physics 2009, 75 (4), 1111-1118. DOI: 10.1016/j.ijrobp.2008.12.055

-----------------------------------------------------------------------------------------

Second report response: 

Dear reviewer,

We appreciate the thoughtful and thorough feedback you provided. We have made the following necessary changes to our manuscript as a result of your feedback:

  1. I previously emphasized to the authors that they were mis-using the term "optic tract" and suggested that they either use the term "anterior visual pathway" or specify retina, optic nerve, etc. They appear to have done so in some areas of the Discussion but not in all and not in the Introduction, in which the term "optic tract" continues to appear multiple times.Please revise.

Response author: we have revised the term “ optic tract” to retina, optic nerve and chiasm in the introduction.

  1. Page 3, first paragraph, second line. The authors use the phrase "retina, chiasm, and optic nerve"; it would be more appropriate to change this to "retina, optic nerve, and chiasm"

Response author: we have changed the order of the terms at page 3 first paragraph as suggested by the reviewer.

  1. Page 5, first paragraph (section 3.1), first sentence, please add a comma after the word "total"

Response author: we have added a comma after the word “total” at page 5, first paragraph.

  1. Page 5, first paragraph, second sentence, please add a comma after the word "studies"

Response author: we have added a comma after the word “ studies” at page 5, first paragraph.

  1. Page 5, first paragraph, second sentence. I do not understand the phrase "cavernosus tumor". Do you mean a tumor of the cavernous sinus or are you referring to a cavernous malformation? Please clarify.

Response author: Indeed, this should be “ cavernous malformation” not “ cavernosus tumor”. We have  corrected this at page 5, first paragraph, second sentence.

  1. Page 5, first paragraph, line 10. Please delete the word "only". Let the reader decide if the numbers of studies are "significant"

Response author: the word “ only” has been deleted.

  1. Page 5, last paragraph, first sentence. Please add a comma after the word "total".

Response author: a comma is added after the word “ total” at page 5, last paragraph, first sentence.

  1. Page 5, last paragraph, second sentence. Please change the word "From" to "In".

Response author: we have changed the word “ from” to “ in” at page 5, last paragraph, second sentence.

  1. Page 6, first paragraph. The word "meningioma" is mis-spelled twice (meningeoma" and the word "cavernosus" should be changed to "cavernous.

Response author: both words“ meningioma” and “ cavernous” are now spelled correctly at page 6, first paragraph.

  1. Page 8, first paragraph, first sentence. This sentence does not make sense: "Information concerning the doses received by the retina, optic nerve, and chiasm were limited reported in the included  studies." I assume you mean to say that in the studies you reviewed, there was limitation information regarding the doses to these structures? Please rephrase the sentence.

Response author: We have rephrased the sentence with “ The studies included in the analysis reported limited information on the doses received by the retina, optic nerve and chiasm”.

  1. Page 8, first paragraph, fourth sentence. Please change the phrase "nerve, chiasm, and/or retina" to "retina, nerve, and/or chiasm". Similarly, change the last sentence so that retina comes first, then nerve and then chiasm.

Response author: the order of the phrase at page 8, first paragraph, fourth and last sentence has been changed to “retina, optic nerve and chiasm”

  1. Page 8, second paragraph (section 3.3). Last sentence, please change the word "since" to "as"

Response author: the word “ since” has been changed to “ as” at page 8, second paragraph.

  1. Page 8, Discussion. As noted in #1, the term "optic tract" appears several times. Please change!

Response author ; the term “ optic tract” in the discussion has been revised as suggested in #1.

  1. Page 9, Discussion, first paragraph, line 9. Please change the word "they" to "the authors".

Response author: the word “ they” has been changed to “ the authors” at page 9, discussion, first paragraph, line 9.

  1. It is interesting that although the dose to the retina that causes radiation-induced retinopathy is dose-related, with virtually no cases when the dose is <50Gy, there remain cases (<2%) of radiation-induced optic neuropathy that occur with doses <50 Gy. I believe the authors should address this issue.

Response author:  to address this issue we have added the following text in line 253-262 “ This review reports that cases of  radiation retinopathy is rare at doses below 50Gy while radiation optic neuropathy can occur at doses less  <50Gy. The optic nerve is a critical structure with a limited ability to repair itself. [1] Of the studies reporting neuropathy at doses less then 50Gy consist of pituitary tumors 59%. The remaining 41% of the studies consist of cavernous sinus meningioma and craniopharyngioma. The most common neuro-ophthalmological symptom of pituitary tumors prior to radiation therapy is impaired vision. This is caused by compression of the optic chiasm and the visual defects depends on the degree and site of optic nerve compression[2]. If there has been pre-existing factors such as  compression and dysfunction of the optic nerve prior to radiation therapy this may contribute to increase the susceptibility of radiation neuropathy after  radiation therapy and hence explain the rare cases <50Gy.

  1. Page 9, second paragraph, line 14. The authors indicate that the assessment of the fundus is by a slit lamp fundus examination. Although it is true that many ophthalmologists use a slit lamp with a hand-held contact lens for assessment of the fundus, others use an indirect or even a direct ophthalmoscope. I'm not sure that the authors should make this so specific. The key is the examinatoin of the fundi, not what instrument you use to examine them.

Response author: we have deleted the “ slit lamp” and replaced it with examination of the fundus”

  1. Discussion, page 10, The authors emphasize the imaging tools (ie, OCT, OCTA, UWFA) used to improve recognition of retinopathy and optic neuropathy; however, these tools only identify structural abnormalities and not functional ones. They should add something about the use of electrophysiology (full-field ERG, PERG, VEP) to identify evidence of dysfunction, particularly with respect to radiation-induced optic neuropathy.

Response author: we have addresses this in line 263-267 with the following text

“  Next to the assessment of the structural abnormalities also the assessment of functional abnormalities could contribute to this. Electrophysiology tests, such as full field electroretinography, pattern electroretinography and visual evoked potential can provide valuable information about the functional status of the retina and optic nerve.”

  1. Page 10, second paragraph, sentence 3rd from the end. The authors use the abbreviation "WFA". Do they really mean WFA or should this be UWFA, as that is what they have been discussing in the paragraph?

Response author: indeed this should be abbreviated with “ UWFA” and not “ WFA”. 

  1. Page 10, last paragraph, last sentence. Please change the word "Since" to "As" or "Because"

Response author: the word “ Since” has been changed to “ as”.

  1. Page 11, last paragraph, last sentence. Given that radiation-induced optic neuropathy can occur (rarely) with doses less than 50 Gy, doesn't it make sense for all patients to be screened for optic neuropathy, regardless of the dose? Or at least say that patients should be warned that because radiation-induced optic neuropathy can occur at doses less than 50 Gy (1.7% is still significant if you are one of the 1.7%!), any patient who develops decreased vision after radiation therapy that encompasses the optic nerve, regardless of the dose, should be told to see an ophthalmologist immediately

Response author: we have added  the last sentence of the conclusion with the following text in line 384-387

Patients who experience reduced vision following radiation therapy involving the optic nerve should be advised to seek consultation with an ophthalmologist, irrespective of the radiation dose received.”

We hope that the revision we have made address the concerns raised by the reviewer. We look forward to the opportunity to have our manuscript considered for publication.

Kind regards,

Buket Kinaci- Tas

References

  1. van den Bergh, A. C.; Schoorl, M. A.; Dullaart, R. P.; van der Vliet, A. M.; Szabo, B. G.; ter Weeme, C. A.; Pott, J. W. Lack of radiation optic neuropathy in 72 patients treated for pituitary adenoma. Journal of neuro-ophthalmology : the official journal of the North American Neuro-Ophthalmology Society 2004, 24 (3), 200-205. DOI: 10.1097/00041327-200409000-00003
  2. Drummond, J. B.; Ribeiro-Oliveira, A., Jr.; Soares, B. S. Non-Functioning Pituitary Adenomas. In Endotext, Feingold, K. R., Anawalt, B., Blackman, M. R., Boyce, A., Chrousos, G., Corpas, E., de Herder, W. W., Dhatariya, K., Hofland, J., Dungan, K., et al. Eds.; 2000.

Reviewer 3 Report

In Result 3.2 you stated: "Optic neuropathy prevalence was observed at prescribed doses of > 50Gy and <50Gy at 4.5 and 1.7%, respectively. This difference was significant (p=0.001) (Figure 6).". But, in conclusion, you stated, "Ophthalmologic evaluation is recommended for patients receiving >60Gy to their optic nerve". Why these two dose constraints are different?

Author Response

Reviewer 3:

Thank you for your additional comments and for emphasizing the probabability of optic neuropathy at doses higher than 50Gy compared to doses lower than 50Gy.

  1. In Result 3.2 you stated: "Optic neuropathy prevalence was observed at prescribed doses of > 50Gy and <50Gy at 4.5 and 1.7%, respectively. This difference was significant (p=0.001) (Figure 6).". But, in conclusion, you stated, "Ophthalmologic evaluation is recommended for patients receiving >60Gy to their optic nerve". Why these two dose constraints are different?

Response author:

Our intent with the addition to the conclusion was to emphasize the importance of seeking consultation with an ophthalmologist if a patient experiences reduced  vision following radiation therapy involving the optic nerve, regardless of the radiation dose received. We believe that it may not be feasible or necessary to send every patient who received doses lower than 50Gy for ophthalmological examination. Since the probability is less compared to doses higher than 50Gy.  We have add the conclusion with the following text in line 384-387:

 Patients who experience reduced vision following radiation therapy involving the optic nerve should be advised to seek consultation with an ophthalmologist, irrespective of the radiation dose received”

We hope that this addition clarifies our opinion on ophthalmological examination and thank you again for your insightful comments.

Kind regards,

Buket Kinaci- Tas

Round 3

Reviewer 1 Report

The authors have adequately addressed my comments and recommendations. There remain some minor grammatical and syntax errors that need to be addressed, particularly in the several sentences they added (at my request) in the Discussion on radiation doses that cause retina vs optic nerve damage.